# Improving the Therapeutic Relationship When Prescribing Antidepressants: A Pilot Study

**DOI:** 10.3390/healthcare11212825

**Published:** 2023-10-25

**Authors:** Konrad Michel, Daniela Lutz-Beck, Sylva Engeroff

**Affiliations:** 1University Hospital of Psychiatry, University of Bern, CH-3008 Bern, Switzerland; 2Independent Researcher, CH-3007 Bern, Switzerland; 3Independent Researcher, CH-3013 Bern, Switzerland

**Keywords:** pharmacotherapy, antidepressants, doctor–patient relationship, therapeutic relationship

## Abstract

Studies indicate that the quality of the doctor–patient relationship moderates the effect of pharmacotherapy. To enhance the quality of the therapeutic relationship in the pharmacotherapy of depression, we developed a brief manual with interactive materials for residents in psychiatry and their patients. In a pilot study at a psychiatric university hospital’s outpatient department, we compared patient-centered treatment parameters of a first patient group treated as usual and a second patient group treated using the manual. The study had no influence on the choice of medication. In the manual group, patient satisfaction with the doctor–patient relationship increased significantly at the three-month follow-up. Depression parameters declined in both groups, without group differences. Continuation of antidepressant medication at six months was higher in the manual group. In conclusion, a simple intervention using written materials for doctors prescribing antidepressants improved doctors’ and patients’ satisfaction with treatment.

## 1. Introduction

Depression is one of the most common psychiatric disorders [1]. The effect of antidepressant therapy depends to a large extent on patient compliance in consistently taking prescribed medication for an adequate length of time. However, thirty to fifty percent of patients discontinue antidepressant therapy before the end of three months [2,3], often without informing prescribing physicians [4]. Barriers to treatment adherence are numerous, including problems with side effects and nonresponse to pharmacotherapy that require further treatment steps, which may strain a patient’s motivation [5,6]. Furthermore, patient knowledge about depression and antidepressant medication is often incorrect and misleading [7].

Studies indicate that treatment engagement and outcome are moderated by the quality of the clinician–patient interaction in psychotherapy as well as in pharmacotherapy [8,9,10]. Ideally, a therapeutic relationship between a physician and a patient requires a collaborative, trust-building approach on the part of the therapist. It has been argued that a patient’s expectation and hope to receive help are essential for effective treatment [11,12,13]. This finding is supported by studies using imaging methods. For instance, patients treated with fluoxetine or a placebo, who in the course of treatment were found to be responders, in both conditions showed significant changes in neuronal activity in the cingulate gyrus at one week of treatment, well before clinical improvement [14,15]. Therefore, it can be argued that an essential task of the prescribing doctor is to initiate adaptive neural processes instrumental for improvement, a process that, to a large extent, depends on the health professionals’ skills to build a working alliance [16,17]. Recommendations for patient-centered prescribing of antidepressants include easy to understand disease models [18,19], a clear treatment framework, written material about diagnosis and treatment, exploration of patients’ attitudes towards pharmacotherapy and past experiences with drugs, collaborative psychoeducation, and shared decision making [20,21,22].

This pilot study was conducted at a university psychiatric hospital’s outpatient clinic for depression treatment. Patients were referred by general practitioners for clinical evaluation and indication of antidepressant treatment. After antidepressant medication was established, patients were referred back to their general practitioners. This special clinic receives approximately 100 referrals per year. To improve psychiatry residents’ clinical skills while working in this service, we developed written materials aimed at enhancing the quality of the doctor–patient relationship in prescribing antidepressants. This study project was a doctoral thesis, primarily practice-oriented, with a main focus on the quality of the doctor–patient relationship. The manual was developed by the three authors and issued to residents as a folder to be used for each new patient. Materials for interactive use included basic information about the therapeutic relationship, checklists for psychiatrists, and handouts for patients. The latter included easy-to-understand psychoeducational information that explained causal factors of depression, antidepressants mode of action, and the latency of clinical improvement. In each session, changes in depression symptoms and side effects during follow-up were assessed and recorded collaboratively with the patients.

Furthermore, at the end of the study, residents were asked to provide feedback about the manual’s usefulness.

We hypothesized that patients in the manual group would indicate higher satisfaction with the therapeutic relationship and the treatment-relevant information received. As a secondary outcome measure, we expected that this patient group would show better treatment compliance and more clinical improvement during the follow-up period compared to the TAU group.

## 2. Materials and Methods

### 2.1. Study Design

Ten psychiatry residents working in a university psychiatric department’s special outpatient clinic for antidepressant pharmacotherapy agreed to participate in this pilot study (Figure 1).

In the study’s first phase, residents saw a group of patients referred for a first-time antidepressant treatment or a change in antidepressant medication, according to the institution’s standard clinical practice. Residents performed clinical evaluations, and under the supervision of a senior psychiatrist, decided on antidepressant medication. Antidepressants used included SSRIs, SNRIs, and tricyclics.

In the study’s second phase, during a one-hour meeting, the same residents were informed about the rationale of a manual designed to improve the therapeutic relationship in depression pharmacotherapy. They were invited to use the interactive materials to treat a second group of patients. Patients were informed about this study verbally and in writing at initial contact. They provided written consent for this study’s use of anonymized data, and to be contacted by the study team during a six-month follow-up. Patients were informed that this study was for quality assurance and treatment improvement and that it had no influence on the choice of antidepressant medication. Because of this study’s design, application to the local Ethics Review Committee for this pilot study was waived.

### 2.2. Questionnaires

During the first session, data from hospital records, which included age, sex, age at the onset of disease, number of previous episodes of disease, education, and clinical diagnosis, were collected. Levels of depression were determined using the Beck Depression Inventory [23]. Immediately following the first appointment, patients completed the Helping Alliance Questionnaire (HAQ) [24] and the Antidepressant Compliance Questionnaire (ADCQ) [19]. The HAQ assesses the patient-rated quality of the therapeutic relationship during psychotherapeutic treatment. Bassler et al. [25] propose two subscales: relationship satisfaction (6 items) and treatment satisfaction (4 items). The first subscale includes questions regarding the affective quality of the therapeutic relationship and trust in the therapist; the second subscale refers to the patient’s satisfaction with the treatment outcome. The Antidepressant Compliance Questionnaire (ADCQ) focuses on patients’ attitudes and beliefs about depression and antidepressant treatment, and their satisfaction with the information received. The questionnaire includes 33 items; answers are provided using a 4-point Likert scale. The authors’ German translation of the questionnaire was used, after translation into English and matching with the original. 

At the end of the third session, and three months after the initial session, patients completed the HAQ, the ADCQ, and the BDI. When treatment was continued elsewhere, study documentation was sent to the practitioners responsible for continued treatment. Information regarding medication intake was collected at 6 months follow-up through phone calls to patients.

### 2.3. Participants

This study’s psychiatry residents included 5 females and 5 males with an average age of 36 (32–40). They had an average of 5.8 years of clinical experience after graduation (2.5–10 years), and 4.5 years (0.1–7 years) in psychiatry.

Patients referred to the special clinic were included in this study if they fulfilled the following inclusion criteria: age 18 to 65, unipolar depression (ICD-10: F32.x or F33.x), and indication for a first-time antidepressant medication or a change to another antidepressant. Exclusion criteria included current substance use other than nicotine, bipolar affective disorder (ICD-10: F31, and psychotic disorders (ICD-10: F20–F29).

### 2.4. Manual

Resident psychiatrists received folders with 11 inserted sheets for interactive use with each new patient, focused on:(1)General information for residents about the therapeutic relationship concept.(2)Patient attitudes towards medication; the manual included example questions to be used during the first patient encounter (“have you been taking antidepressants before? If yes, what was the name of the drug? Did it help? Did you have side-effects? What is your opinion about starting on an antidepressant now?”). Residents recorded patients’ answers in writing.(3)Models of depression, including six sheets for interactive use about depression risk factors and psychological and biological models of depression (the latter with a graphic model of the HPA axis and an easy-to-understand description of the glucocorticoid receptor hypothesis), and a graphic illustration of step-by-step recovery from depression, including the so-called Kupfer schema [26]. The sheets were designed to be completed collaboratively and given to patients to take home.(4)In each session, doctors and patients collaboratively completed a 17-item depression questionnaire based on the F32, ICD-10 symptomatology.

### 2.5. Statistical Analysis

Data analysis was performed using the SPSS program (IBM SPSS Statistics Version 20). Group differences at t1 (start of treatment), t2 (third session), and t3 (three-month follow-up) were calculated using t-tests for dependent samples. Chi-square tests were used to compare antidepressant continuation at 6 months. 

## 3. Results

### 3.1. Patient Sample

Thirty-five patients were recruited for the control group, and 22 for the manual group. Due to dropouts, the number of remaining patients with complete t1 to t3 datasets were 24 and 16, respectively. This sample was used for follow-up analysis. The two groups did not differ in terms of sex (control group 12m/12f, manual group 7m/9f), age (37.20 (SD 14.01) vs. 37.78 (SD 12.09), respectively), or depression severity at the beginning of treatment. The initial mean BDI scores were 28.68 (SD 10.41) in the control group and 26.49 (SD 9.27) in the manual group (see Table 1 below).

### 3.2. Questionnaires

*Helping Alliance Questionnaire (HAQ*). At t1, the groups’ mean HAQ scores did not differ (control 26.2 vs. manual 25.4). At t2, the control group’s mean HAQ dropped to 22.5, whereas the manual group’s mean HAQ significantly increased to 27.9 (SD 2.81, t = 1.97, df = 23, *p* = 0.034). A similar pattern was found in the “relationship satisfaction” subscale. At t3, the manual group’s mean HAQ decreased again to 25.9 (control 23.9).

*Antidepressant Compliance Questionnaire (ADCQ).* There were no significant differences between the mean values of the groups’ total scores. The manual group’s mean increased from t1 to t2 (101.2 to 103.6), while the control group’s mean did not change (102.2 and 102.7, respectively). Similar to the HAQ responses, the manual group’s subscale “physician-patient relationship” increased from t1 to t2, then decreased from t2 to t3. In response to questions about the provision of information (items 16–24), the manual group indicated that they were better informed at all three time points.

*Beck Depression Inventory (BDI*). Both groups’ scores decreased significantly from t1 to t2 (t = 4.97, df = 26, *p* < 0.001), as well as from t1 to t3 (t = 5,54, df = 26, *p* < 0.001), with no significant group differences.

*Continuation of drug treatment.* At six-month follow-ups, 14 of 16 patients in the manual group reported that they were still taking the prescribed medication; in the control group, only 2 of 24 patients reported that they were still taking antidepressant medication (chi-square: *p* < 0.005).

### 3.3. Psychiatrists’ Feedback about the Manual

Psychiatrists’ feedback regarding using the manual was consistently positive. Most stated that they liked the manual because it helped them explain the nature of depressive disorders, work collaboratively with patients, and motivate patients for antidepressant treatment. Some residents said they particularly liked that they were free to choose materials individually with each patient. Others stated that the manual gave them more confidence in dealing with depressed patients. 

## 4. Discussion

Published recommendations for prescribing antidepressants mainly focus on drug selection algorithms; only a few include patient-centered psychoeducational recommendations or checklists aimed at fostering the therapeutic relationship in pharmacotherapy [21]. Our pilot study examined the effect of a manual aimed at improving the therapeutic relationship and its acceptance by resident psychiatrists. In our study’s first phase, a control patient group received treatment as usual; in its second phase, after a short introduction by the authors, psychiatrists were encouraged to treat patients (manual group) using the interactive manual. In the third session, patients in the manual group showed a significant increase in satisfaction with the therapeutic relationship (HAQ); in contrast, the control group’s HAQ mean scores decreased from t1 to t2. We interpreted this as a sign of an increasing working alliance resulting from the residents’ manual-based approach. At three months follow-up, when most patients had been discharged back to their referring practitioners, HAQ scores declined. This indicates that patients correctly related the questionnaire to the actual health professional responsible for the follow-up treatment. The difference in HAQ scores was expected, considering that referring practitioners were not familiar with the manual’s content.

Regarding knowledge of antidepressant medication provided (ADCQ), the manual group’s values were consistently higher, but the difference did not reach statistical significance. Demyttenaere et al. [22] showed that better knowledge increases treatment engagement and the likelihood that patients inform their doctors about problems with medication, which is essential to monitor and adjust treatment [9]. A positive finding was that at six-month follow-up, most manual group patients were still taking their prescribed medication; in contrast, most control group patients had stopped taking antidepressants. We assume that the psychoeducational material handed out to patients, which included guidelines for the minimum length of antidepressant treatment, had a long-term effect on patient compliance.

Both HAQ and ADCQ results supported our hypothesis that the manual-based interactive approach would improve doctor–patient relationships during outpatient clinic antidepressant treatment sessions.

The hypothesis that a good therapeutic relationship would be associated with a better treatment outcome was not supported by the results. The mean BDI values declined in both groups from the moderate depression range to mild depression. Similarly, Bermejo, Schneider [27] reported that using treatment guidelines was not associated with treatment outcomes. 

Resident’s feedback regarding using the manual was positive, and most of them stated that they will continue to use the interactive materials. Above all, the structured patient-centered and interactive approach was highly appreciated. Unfortunately, we do not know if this exercise in improving the doctor–patient relationship had a long-term effect on the clinical skills of the participating psychiatrists in training.

## 5. Limitations

This pilot study has clear limitations. First, the intervention was minimal, in that psychiatrists used the manual after a brief introduction, with no formal training and no continuous manual-related supervision. The small number of cases did not allow patient differentiation according to depression severity or the number of previous depressive episodes. The manual-based approach was time-limited, usually with three to four sessions. After being prescribed antidepressant medication, patients were discharged from the institution and we could not assess the long-term effects of this outpatient intervention. Future studies should include follow-up interviews using a qualitative approach. Regarding compliance, determining drug levels in the blood would be useful to reliably record drug intake. Finally, we are aware that the Helping Alliance Questionnaire (HAQ) was originally developed for psychotherapeutic treatments and is, therefore, not an ideal measure of the doctor–patient relationship in pharmacotherapy. However, we believe that the principles of a working alliance in the pharmacotherapy of depression do not differ significantly from those of psychotherapy.

## 6. Conclusions

The results of this pilot study indicate that psychiatry residents’ use of the manual aimed to improve the working alliance when prescribing antidepressant medication was associated with greater patient satisfaction with the treatment received. The quality of the therapeutic relationship in this time-limited intervention did not have a direct effect on the course of depression within three months, but at the six-month follow-up, the patient compliance with continuing medication was significantly better in the manual group. The findings suggest that written guidelines for patient-centered and interactive pharmacotherapy can improve the prescribing doctors’ skills in establishing a collaborative therapeutic relationship with their patients.

Note: The manual with interactive materials (in German) is available as a pdf from the author upon request.

## Figures and Tables

**Figure 1 healthcare-11-02825-f001:**
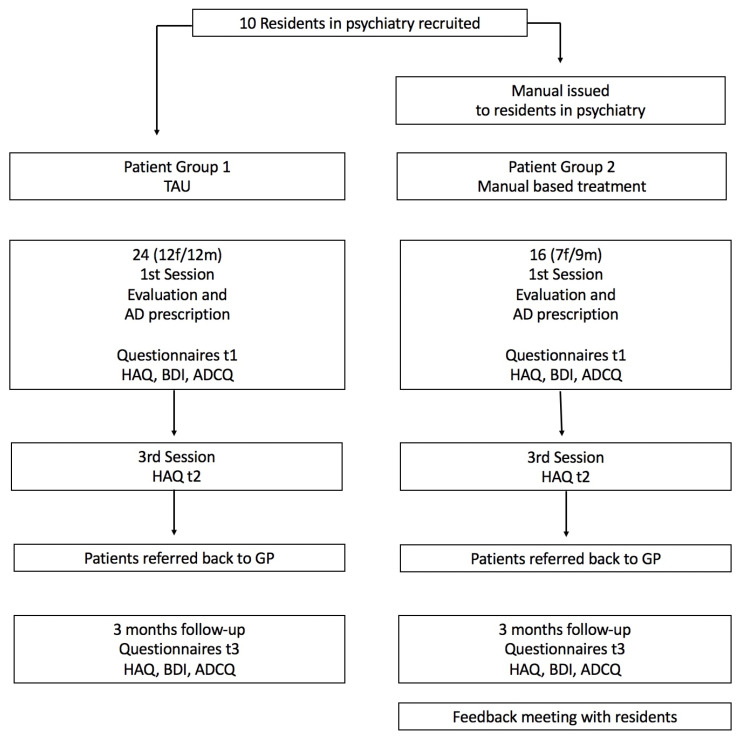
Study flow chart. Caption: Study flow chart. The numbers of patients included (24 and 16, respectively) are the cases with complete datasets: t1 (1st session), t2 (3rd session), t3 (3 months follow-up).

**Table 1 healthcare-11-02825-t001:** Mean scores and standard deviation, t1–t3. Caption: HAQ, Helping Alliance Questionnaire; ADCQ, Antidepressant Compliance Questionnaire; BDI, Beck Depression Inventory; t1 (1st session), t2 (3rd session), t3 (3 months follow-up).

	Control N = 24			Manual N = 16		
	t1	t2	t3	t1	t2	t3
**HAQ**	26.23 (5.24)	22.50 (5.65)	23.90 (6.58)	25.40 (5.92)	27.91 (10.60)	25.92 (10.74)
**ADCQ**	102.26 (10.39)	102.7 (10.05)	103.08 (9.71)	101.21 (10.55)	103.63 (13.81)	100.51 (7.98)
**BDI**	28.68 (10.41)	16.11 (11.34)	13.98 (12.85)	26.49 (9.27)	15.91 (12.72)	15.12 (10.41)

## Data Availability

Research data is available from K.M.

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
