# Peer review of "Improving the Therapeutic Relationship When Prescribing Antidepressants: A Pilot Study"

_healthcare, 2023, doi:10.3390/healthcare11212825_

Round 1

Reviewer 1 Report

Method section

Please draw algorithm of your intervention for better understanding of the study.

Discuss “standard clinical practice of the institution”, which medication is commonly used? Please bring a brief discussion

In the result section:

Please bring the baseline characteristics and main results in table or figure.

 Overall, i dot 

Author Response

See attachment (Word)

Reviewer 2 Report

Some specific comments are as follows.

1. The text does not reflect the characteristics of the manual, the manual description is too simple.

2. The sample size is too small, the analysis data is insufficient, and there is a lack of charts.

3. It is recommended that more in-depth work be done.

English should be improved.

Author Response

See attachment (Word)

Reviewer 3 Report

Thank you for the opportunity to review this manuscript. The overall goal of the study is both interesting and important. However, several comments and suggestions should be addressed.

1) Serval different terms are used to address the relationship between doctor and patient, e.g., therapeutic relationship, pharmacotherapeutic relationship, doctor-patient relationship, therapeutic alliance, doctor-patient alliance, therapeutic working relationship, doctor-patient working alliance. Although the variety of related terms/constructs is a well-known problem in the field, the use of a consistent terminology would be recommended for the sake of clarity. Otherwise, the theoretical distinction between these terms should be addressed.

2) It should be mentioned why a pilot study and not an RCT or CT was conducted.

3) Please give a short description of the department where the study was conducted (inclusion criteria, average length of stay, number of patients treated annually etc.).

4) It should be discussed if the results could have been confounded by seasonality or changes in the team, as the intervention and control group were not parallel

5) Information on the length of hospital stay of both groups – another potential confounder - is necessary

6) It is unclear why the results on the therapeutic relationship are reported first in the result section, although no hypothesis was explicitly introduced. Either these results should be reported after the main results (i.e., treatment engagement and depression), or hypotheses on therapeutic relationships should be explicitly formulated in the introduction

7) Please specify which standardized questionnaire was used for the assessment of the psychiatrist’s feedback (line 96/97)

8) The results on the differences between the two groups should be systematically reported in a table (including effect sizes) for all relevant outcomes.

9) It is not reported whether recurrent depressive episodes (F33) or patients with switch of antidepressant medication, both potential confounders, were more frequent in one of the two groups. Please add this information

10) line 86: Please specify when the questionnaires were completed at 3 and 6 months or at the end of treatment

Author Response

See attached Word document

Reviewer 4 Report

This manuscript presents a pilot study that investigates the impact of an interactive manual on the therapeutic relationship between residents in psychiatry and patients undergoing antidepressant therapy, addressing a critical gap in existing antidepressant prescribing recommendations that often overlook patient-centered psychoeducation. The study's well-structured design includes a control group and a manual group to assess the manual's influence. The results demonstrate a significant enhancement in the satisfaction of the therapeutic relationship in the manual group, measured by the HAQ, in contrast to a decline in the control group, highlighting a positive impact on the doctor-patient alliance. Additionally, the manuscript reports higher levels of patient knowledge and more positive attitudes toward antidepressant medication in the manual group, resulting in improved compliance—an encouraging finding for patient care.

I am curious to see more patients included and more resident physicians using it and compiling the results. At this stage, the publication is appropriate, but the sample is size is very limited..

Author Response

See attached Word document

Round 2

Reviewer 1 Report

Dear Authors,

Thanks for your response.

Author Response

Thank you for the positive answer!

Reviewer 2 Report

This manuscript has been improved in logic and detail after modification, but there are still some small problems to be improved. 

1. The figures in lines 140-142 can be appropriately placed in the corresponding text. 

2.What is t1-t3 in line 142, should be marked when it first appears.

Author Response

Thank your for these 2 important points made. The Fig.1 has now been moved to the section "Study design", where indeed, it belongs!

In the caption we have now included the details of t1 - t3:  t1 (1st session), t2 (3rd session), t3 (3 months follow-up).

Reviewer 3 Report

The author(s) addressed the issues raised by the reviewers well, and I endorse publication of this manuscript.

Author Response

Thank you for the ok!